# Analyzing the Magnesium (Mg) Number of Olivine on the Lunar Surface and Its Geological Significance

**Chao Zhou [1,2]** , **Yuanzhi Zhang [3,4,*]** , **Shengbo Chen [2]** and **Bingxue Zhu [2]**

[1]   Key laboratory for Ecological Environment in Coastal Areas (SOA), National Marine Environmental Monitoring Center, Dalian 116023, China

[2]   College of Geo-exploration Science and Technology, Jilin University, Changchun 130026, China

[3]   Key Laboratory of Lunar and Deep-Space Exploration, National Astronomical Observatories, Chinese Academy of Sciences, Beijing 100101, China

[4]   School of Astronomy and Space Science, University of Chinese Academy of Sciences, Beijing 100049, China

*   Correspondence: zhangyz@nao.cas.cn

**Abstract:** Olivine formation is directly related to Mg/Fe content. It is also significant in estimating the geological evolution of the moon. In this study, an estimation model of relative Mg number (Fo#) for lunar olivine was presented through multiple linear regression statistics. Sinus Iridum, the Copernicus Crater, and the pyroclastic deposit in the volcanic vents in the southeast of Orientale Basin were selected as the study areas. Olivine distribution was surveyed, and the relative Fo# calculation of olivine was implemented based on Moon Mineralogy Mapper ($M^3$) data. Results demonstrated that olivine in the crater wall of Sinus Iridum and the Copernicus Crater had relatively high Fo#, which reflected the primitive melt. However, the difference in olivine spectral features between Sinus Iridum and the Copernicus Crater indicated different crystallization modes. The olivine in the pyroclastic deposit in the volcanic vents in the southwest of Orientale Basin also presented high Fo#, which indicated that the olivine was formed via rapid cooling crystallization and was accompanied by volcanic glass substances. As a result, the olivine relative Fo# calculated from the estimation model exhibited an important constraint implication for explanation of its causes.

**Keywords:** lunar surface; Mg number; olivine; $M^3$ images; modified Gaussian model

## 1. Introduction

Olivine's Fe and Mg contents are directly correlated with its formation. The compositional variation of olivine is mainly attributed to Fe and Mg metathesis, which is represented by Mg number (Fo#). The olivine formed during the early, rapid-cooling crystallization of magma on the moon is magnesian (high Fo#), whereas the olivine formed during the late, slow crystallization of magma is ferrous (low Fo#) [1,2]. High Fo# crystals, which develop in magma/melt first and take up the Mg, and Mg-rich olivine, diapirically rise because of their relatively low density, resulting in the production of Fe-rich olivine during late stage cooling [3]. Accordingly, studies of olivine distribution on the moon, as well as of the Fe and Mg contents of olivine, can indicate the material composition of the deep sections of the moon and the evolution mechanism of the magmatic source.

A series of explorations on the moon were conducted to obtain various remote sensing data. The spectral resolution of hyperspectral data is sensitive to the diagnostic absorption properties of minerals, which can be used to study the distribution and composition of olivine on the lunar surface using tools, such as the Moon Mineralogy Mapper ($M^3$) [4] onboard the Indian Chandrayaan-1 and the Spectral Profiler (SP), and onboard the Japanese explorer SELENE/Kaguya. Yamamoto et al. [5] screened 34 olivine-rich matching positions using Spectral Profiler data, adopting the absorption characteristics

near 1050 ± 30 nm of olivine as the standard. They determined that the majority of these positions were on the edges of a large-scale impact crater and that the spectral properties of the corresponding olivine were consistent with those of the dunite produced by the lunar mantle. Hence, Yamamoto et al. [5] believed that these olivine-rich substances might predict certain deep-source characteristics of the moon and would significantly restrict explanations of the geological evolution of an impact crater in future studies. Combe et al. [6] extracted the endmember spectra of olivine materials from $M^3$ hyperspectral data using a multiple endmember linear spectral decomposition model. On the basis of $M^3$ hyperspectral data, Qiao et al. [7] produced images of the absorption characteristic parameters of the minerals in Sinus Iridum and recognized olivine in the northern crater wall. They identified three possible sources of olivine in Sinus Iridum. Li et al. [8] conducted a multivariate regression analysis based on the olivine content in Lunar Soil Characterization Consortium (LSCC) samples and on the modified Gaussian parameters of the sample spectrum, then they established an inversion model of the olivine content in Sinus Iridum. The results showed that the olivine content in Sinus Iridum was generally low and was distributed mainly in the southeast. This observation is not consistent with the conclusions of Yamamoto et al. [5] and Qiao et al. [7], that olivine is distributed mainly in impact crater walls.

In view of the importance of mafic mineral's Fo# for determining the cooling history of magma oceans, remote sensing provides strong support for estimates of, or assumption concerning, the Fo# of Mg-suite rocks. Type and abundance of lunar elements can be estimated from gamma-ray spectrometer data, but the reliability of the resulting data suffer from weak Mg signal [9,10]. Lucey and Crites [11,12] used Clementine UVVIS spectra to obtain olivine and pyroxene content, then converted them to their oxides to allow mapping of global Fo# with relatively low spatial resolution using simple stoichiometry. Ohtake et al. reported creating an Fo# map for the lunar highlands based on data from the Kaguya Spectral Profiler. To focus on ferran anorthosite, they used spectral absorption angles between 920 nm and 950 nm to estimate Fo#, avoiding the influence of olivine [9]. When estimating olivine's Fo#, the classic modified Gaussian model (MGM), as proposed by Sunshine et al. [13], calculates the main parameters of the characteristic absorption peak according to substance spectrum, including: Absorption center, full width at half maximum, and absorption peak intensity. Sunshine and Pieters [13] analyzed the spectral features of 1 µm (or 1000 nm) synthetic olivine with different Fe and Mg contents using MGM and confirmed that the MGM method could effectively distinguish the spectra of terrestrial olivine with different Mg and Fe contents. Isaacson et al. [14] implemented the spectral deconvolution of lunar olivine samples and conducted preliminary analysis of the correlation between the spectral features of olivine and Fe/Mg contents. Isaacson et al. [15] also presented a comparative discussion of the Fe and Mg contents of olivine in the Moscoviense impact crater on the moon based on $M^3$ data, and by combining the Fe and Mg content of lunar olivine samples.

A spectroscopic method for estimating olivine composition was put forward by Sunshine et al. [13], after which a revised approach was developed by Isaacson et al. [14,15] which sought to solve complicated issues related to lunar olivine with reference to two factors: Chromite absorption and the effect of continuum slopes. Thus, the composition of lunar olivine can be estimated using a mature algorithm and a hyper-spectral imaging spectrometer. However, more critical attention is still needed to explore the olivine's source and history by means of olivine's Fo# as an important additional clue. Based on the foregoing, the present study applied a prediction model of relative Fo# for lunar olivine using multiple linear regression based on $M^3$ hyper-spectral data. We analyzed the relative Fo# of olivine from the impact craters of Sinus Iridum and Copernicus, as well as from the pyroclastic deposit in the volcanic vents in the southwest of Orientale Basin, along with its proven geological clues, thereby constraining the geologic origin of olivine in the study area.

## 2. Spectral Features of Lunar Olivine

As shown in Figure 1, LR-CMP-14 and LR-CMP-225 are the spectra of lunar olivine at the Apollo 17 landing site. GDS 71.a and GDS 71.b are the spectra of terrestrial olivine from the US Geological Survey spectral library. Typical olivine spectra have three absorption features: A strong absorption near 1050 nm is attributed to $Fe^{2+}$ at the $M_2$ site, while relatively weak absorptions at 850 nm ($M_{1-1}$) and 1250 nm ($M_{1-2}$) are caused by $Fe^{2+}$ at the $M_1$ site [16]. However, lunar olivine often contains Cr-spinel inclusions, which lead to absorptions beyond ~1600 nm [14]. The Fo# of olivine is frequently controlled using the combination of three absorption bands at $M_{1-1}$, $M_2$, and $M_{1-2}$. The decrease in particle size results in increased reflectivity of the spectra of both lunar and terrestrial olivine.

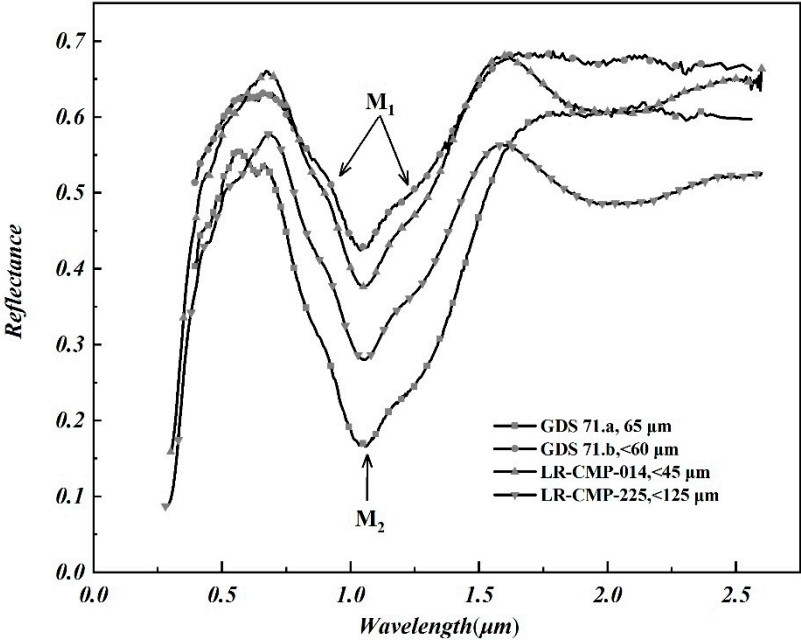

**Figure 1.** Reflectance spectra of terrestrial olivine and lunar olivine.

Olivine-rich samples were found at the landing sites of Apollo 11, 12, 15, and 17. The 15555# sample has the lowest Fo# (approximately 48), whereas the 72415# sample has the highest Fo# (approximately 88). The average Fo# of all the olivine samples is approximately 73. The three absorptions are a function of the olivine's Fo#, and they vary linearly with changing Fo#. Figure 2 shows the measured olivine Fo# of terrestrial and lunar samples compared with the MGM-derived band centers from the research of Sunshine et al. [13] and Isaacson et al. [14,15], respectively. As shown in Figure 2, the absorption band centers at $M_{1-1}$, $M_2$, and $M_{1-2}$ are shifted more toward the longer-wavelength side, and the movements of $M_{1-1}$ and $M_{1-2}$ are more evident than that of $M_2$. This well-characterized feature is the result of the changing distance from metal to oxygen inside olivine, which indicates that trends in lunar olivine should be consistent with those in terrestrial olivine. Isaacson et al. [14,15] have confirmed this view and used these trends to evaluate the robustness of revised MGM for Fo# prediction [14].

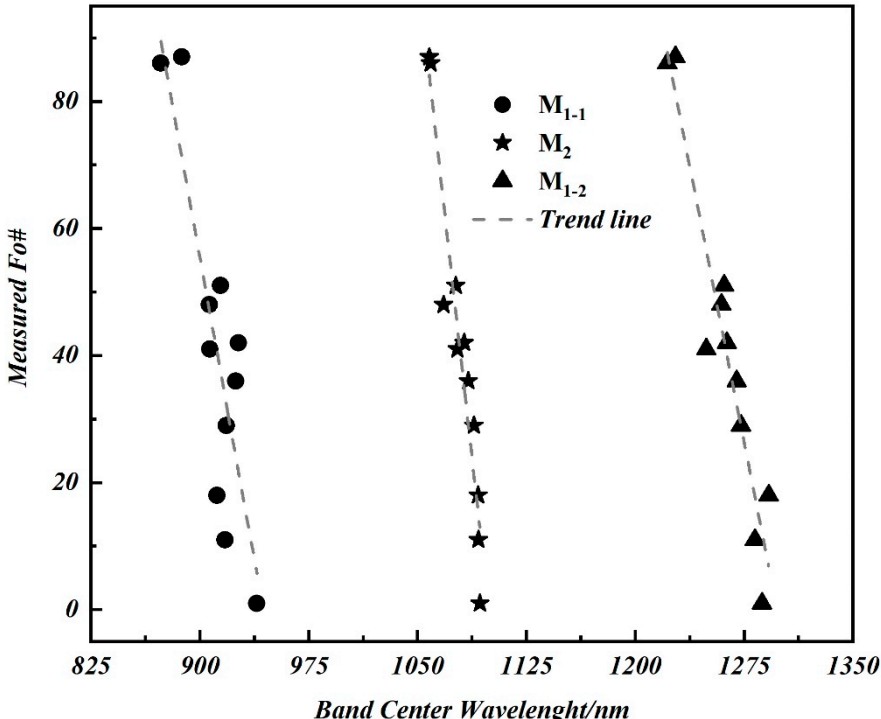

**Figure 2.** The trends between Fo# and of $M_{1-1}$, $M_2$, $M_{1-2}$ of terrestrial and lunar olivine samples adapted from Sunshine and Pieters [13] and Isaacson et al. [14].

## 3. Relative Fo# Calculation Method of Lunar Olivine

$M^3$ is a new-generation lunar exploration hyper-spectral instrument aboard the Indian lunar exploration satellite Chandrayaan-1. Its exploration spectral coverage ranges from visible to near-infrared bands (400 nm–3000 nm). The main scientific purpose of $M^3$ is to identify the major minerals and rocks on the moon [17]. $M^3$ data include global and target exploration modes. This study used L2-level reflectance data of the global pattern with a spatial resolution of 140 m. The number of bands is 85, and the spectral resolution is between 20 nm and 40 nm [18]. In addition, because the current $M^3$ L2-reflectance data show significant thermal residuals at longer wavelengths (i.e., >2500 nm) [19,20], the $M^3$ data need to be cut off at 2500 nm in the subsequent processing.

This study extracted olivine-distributed positions in the study area based on $M^3$ data through the spectral feature fitting (SFF) method, an absorption feature-based methodology using least-squares technique to match unknown spectra to laboratory reflectance spectra, using depth and shape feature [21–24]. The SFF is run to fit for each pixel of $M^3$ data from the olivine spectra of lunar samples, generating a scatter plot of scale and root mean square error (RMSE), which allows selection of areas having relatively high scale and relatively low RMSE, such as olivine-rich regions.

As Figure 2 shows, Fo# exhibits a significant linear relationship with the absorption band centers at $M_{1-1}$, $M_2$, and $M_{1-2}$. However, the prediction of absolute olivine Fo# will require more lunar olivine data to avoid using fitting results from our very limited sample of lunar olivine. Accordingly, we consider that a prediction model for relative Fo# of lunar olivine can be established based on the foregoing trends. First, an equation relating Fo# and three band centers was set up using multiple linear regression, using as data sources MGM-derived band positions and measured Fo# of lunar and terrestrial olivine samples, based on the research results of Sunshine and Isaacson, (as shown in Figure 2). Next, olivine-dominated reflectance spectra selected from $M^3$ imagery were deconvolved using a revised MGM approach developed by Isaacson et al. (2011). Finally, relative Fo# of lunar olivine

was estimated using a fitted equation. As shown in Equation (1), the coefficient of determination for the regression model ($R^2$) is 0.96:

$$Fo\# = 1924.566 - 0.1199\mu_{M_{1-1}} - 0.9244\mu_{M_2} - 0.6149\mu_{M_{1-2}} \tag{1}$$

where $\mu_{M_{1-1}}$, $\mu_{M_2}$, and $\mu_{M_{1-2}}$ are the absorption band centers at $M_{1-1}$, $M_2$, and $M_{1-2}$, respectively, of lunar olivine spectra after MGM decomposition.

To quantify the estimation error of relative Fo# caused by MGM fit, the olivine-dominated reflectance spectra were fit by MGM under randomly selected model initial conditions within the constraint range. Then, combinations of three band centers, randomly generated within the maximum and minimum value of MGM-derived band centers, were used to calculate relative Fo# using Equation (1). Error range is defined as the maximum and minimum values of relative estimated Fo#, which are presented in the form of boxplot. To avoid unreasonable geological-interpretation consequences of the estimation errors, large errors of greater than 20 Fo# units were eliminated.

## 4. Data Processing and Analysis Results

### 4.1. Fo# Analysis of Olivine in Sinus Iridum

Sinus Iridum, an important bay in the northwest of Mare Imbrium, on the lunar nearside, is an important detection zone of the Chang'e satellite series. This study extracted olivine distribution in Sinus Iridum using the SFF method (Figure 3). The base map used digital elevation model (DEM) data from the Lunar Orbiter Laser Altimeter (LOLA) to obtain an intuitive display of the topographic features of the olivine distribution region. According to the geological map of Sinus Iridum [25], clinopyroxenes and orthopyroxenes are the main minerals in Sinus Iridum, along with a little hematite, but olivine is distributed mainly in the walls of some hills at the foot of Montes Jura (e.g., A, B, C, and D in Figure 3). The strata of A, B, C, and D are composed mainly of rock fragments formed by the slumping of material on the slopes and have an average geological age of 3.98 Ga [26,27]. The strata of the crater walls on the north of Sinus Iridum (E and F in Figure 3) consist of complex mixtures of slumped Iridum ejecta and have an average geological age of 4.02 Ga [26,27]. Six spectra were selected from the olivine-rich region as representative spectra. The spectra after continuum removal exhibited distinct olivine absorption features at 1050±30 nm. However, pyroxene was found around the olivine-rich region, which may introduce additional absorption of the olivine-dominated spectra at 1600 nm.

In the olivine-rich regions, more than 10 olivine-dominated spectra were selected for MGM fitting with five Gaussian waveforms. The initial fitting parameters of the Gaussian waveform central value were set at 630 nm (short-wave absorption center), 860 nm ($M_{1-1}$), 1040 nm ($M_2$), 1232 nm ($M_{1-2}$), and 1650 nm (long-wave absorption center), which were the mean fitting results of Isaacson et al. (2010). The relative Fo# of olivine in Sinus Iridum was calculated using Equation (1). As indicated in Figure 4, the error range of predicted relative Fo# in Sinus Iridum is from 9 to 15 Fo# units. Calculating the average value of relative Fo# for each data point, the lowest and highest Fo# are approximately 78 and 86, with an average of 82 and a standard deviation (SD) of 2.4. Compared with the mean Fo# (~73) of the global olivine samples, the olivine in Sinus Iridum generally has a high Fo#, and the geological age of the rich regions is relatively high, which indicates that the olivine in Sinus Iridum resulted from early crystallization during the rapid cooling of lunar mantle crystallization or impact melt. Qiao et al. [7] summarized three ways in which olivine moved to the lunar surface: (1) Lunar mantle olivine, which is exposed on considerable impact, sputtered to the area surrounding the impact crater; (2) olivine moved via direct eruption or overflow during volcanic activities; and (3) magma invaded the superficial layer and then became exposed on impact. Meanwhile, Yamamoto et al. [5] determined that some of larger basins excavated portions of the lunar upper mantle, which has a relatively thin crust to produce the basin-related olivine. However, the inversion results of lunar crustal thickness from high-resolution gravity data also show that the crust of Mare Imbrium is relatively thin (about

10 km) [28]. Accordingly, by combining the results of these analyses, we infer that the majority of olivine in Sinus Iridum was excavated to the surface from the lunar mantle by the later impacts.

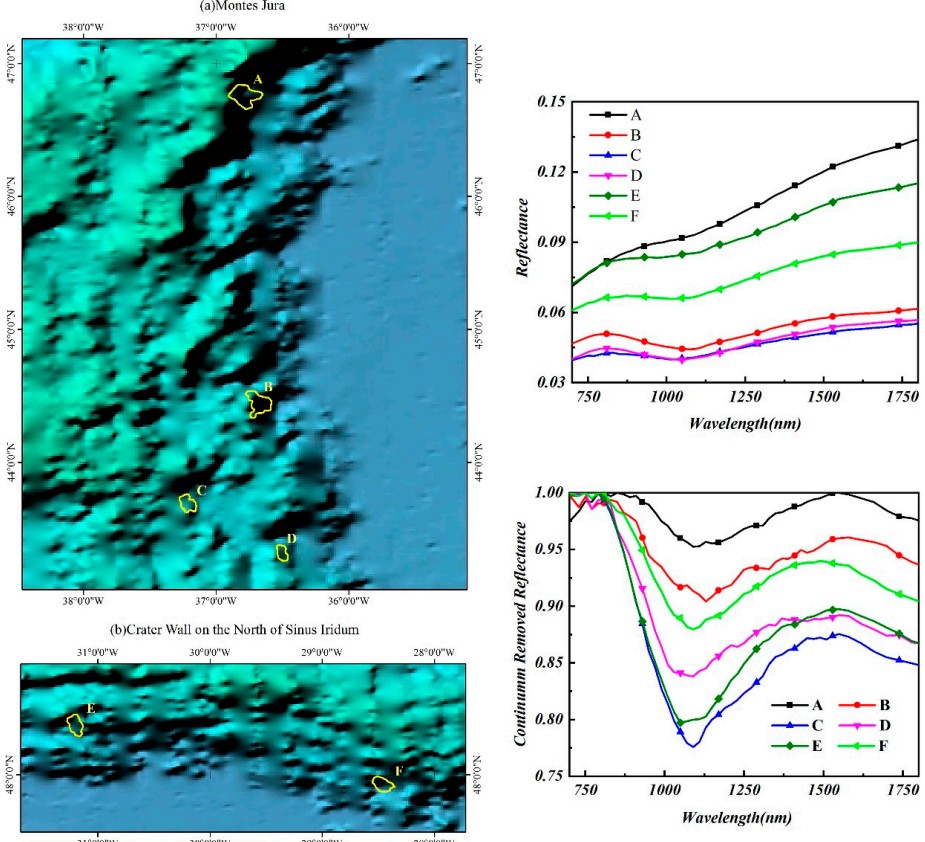

**Figure 3.** Olivine distribution in the Sinus Iridum and its spectra.

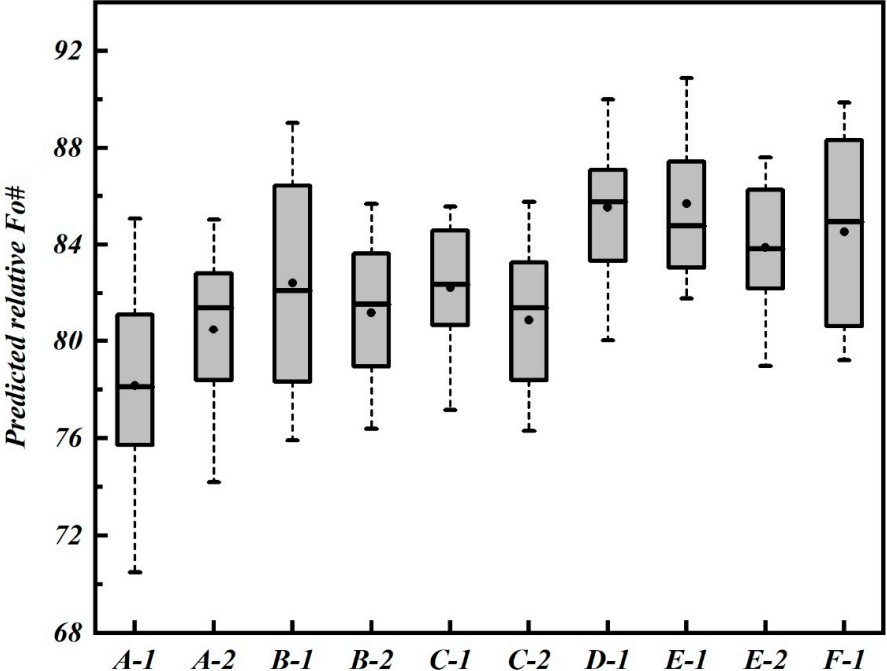

**Figure 4.** Prediction result of relative Fo# for olivine-rich materials in the Sinus Iridum.

### 4.2. Fo# Analysis of Olivine in the Pyroclastic Deposit in the Southeast of Orientale Basin

Orientale Basin is the youngest large-impact basin on the moon, with a geological age of 3.8 Ga [29]. In the southwest of Orientale Basin is a dark ring, which is mainly composed of lunar pyroclastic deposits (LPDs). Head et al. [30] argued that compared with other pyroclastic deposit regions, the thickness of the pyroclastic deposit in the dark ring was relatively thin (<10 m) and was the manifestation of a pyroclastic eruption originating at a fissure vent. Age data revealed that most pyroclastic deposits are of late Imbrium age, generally 3.2 to 3.7 Ga [31]. The LPDs represent material located deeper than the lunar mare basalt, and they are most likely to be representative of primitive lunar magma [32]. In Figure 5, the base map is the reflectivity image of $M^3$ data at 1009 nm. A, B and C in the figure represent the olivine-rich areas, which were distributed mainly in the west of 98° W, clustering in the northwest and southwest corner of the dark ring and the west of the central vent. In the east of 98° W, the minerals in the dark ring are dominated by pyroxene. In addition, a few orthopyroxene–olivine–spinel (OOS) lithological suites are found in the north of the dark ring near the 98° W (as shown in D area of Figure 5) [33].

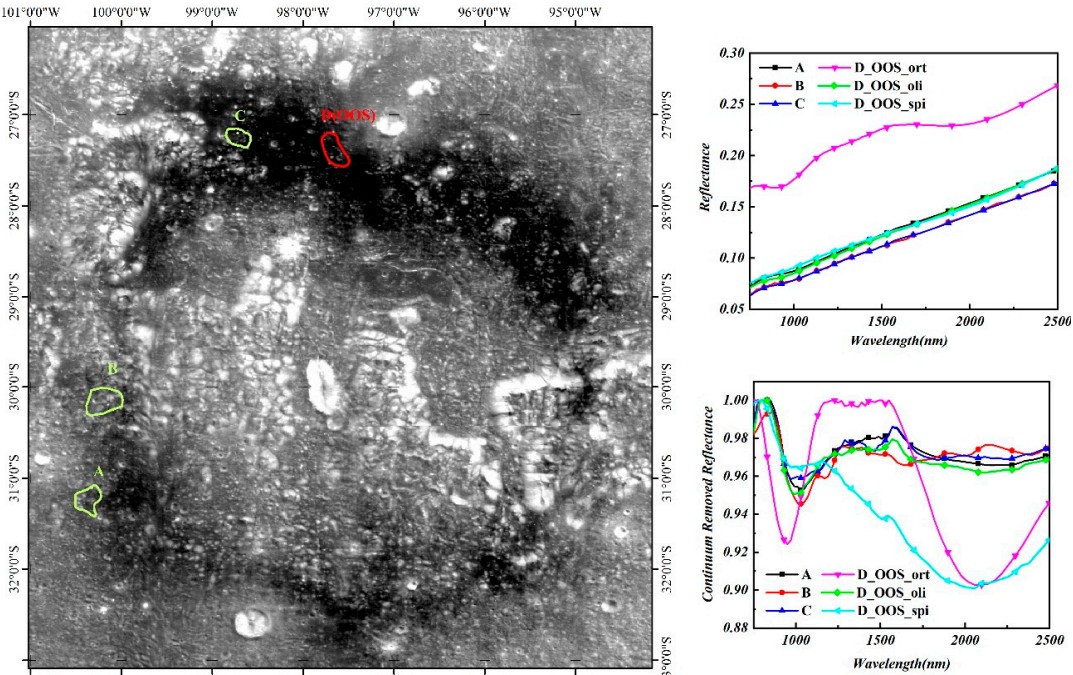

**Figure 5.** Olivine and orthopyroxene–olivine–spinel (OOS) distributions in the dark ring and their corresponding spectra.

In the olivine-rich regions, more than 10 olivine-dominated spectra were selected for MGM fitting and to calculate the relative Fo# of olivine. As shown in Figure 6, the error range of predicted relative Fo# is from 10 to 18 Fo# units. Calculating the average value of relative Fo# for each data point, the lowest and highest Fo# are approximately 68 and 78, with an average of 75 and a SD of 3.5. The relative Fo# of olivine in the dark ring is lower than that of olivine in Sinus Iridum. However, it remains higher than the average Fo# (approximately 73) of all the lunar olivine samples. Lunar Mg-rich olivine represents the rapid early crystallization of magma [32,34]. Magma has a rapid cooling time and produces glassy components. These components are exposed with olivine. Thus, the absorption band center of olivine-dominated spectra in the dark ring moved forward to nearby 1020 nm (particularly region C). Absorption intensity was decreased, whereas absorption peak was widened.

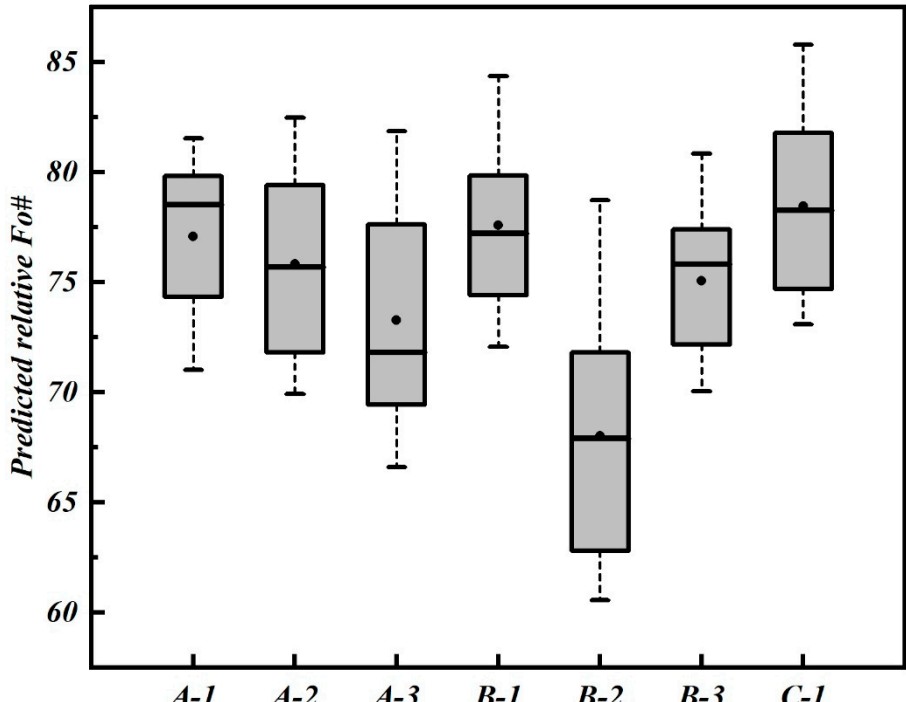

**Figure 6.** Prediction result of relative Fo# for olivine-rich materials in the dark ring.

Two hypotheses have been put forth to explain spinel formation. One suggests that spinel is formed via the stepwise crystallization of magma [35], whereas the other holds that spinel is formed via contact metamorphism [36]. The first hypothesis suggests that magma experiences a long condensation period and produces a large number of crystals. Because this finding contradicts understanding of the causes of Mg-rich olivine, the spinel in the dark ring may have formed via contact metamorphism between magma and surrounding rocks.

### 4.3. Fo# Analysis of Olivine in the Copernicus Crater

The Copernicus Crater is located in the middle-east area of Oceanus Procellarum, close to the lunar equator. It was formed during the late Copernican, and its geological age is approximately 0.8 Ga [37]. Abundant geomorphic features are developed in the impact crater, such as the central peak, ladder-like crater wall, catena, and extensive fusion coating (see Figure 7). The base map is based on DEM data acquired using LOLA.

In the Copernicus Crater, olivine is concentrated mainly in the central peak and the north wall (Figure 7), although some are scattered at the northwest of the crater bottom and in the north crater walls. Compared with the olivine spectra in Sinus Iridum and the dark ring in Orientale Basin, the olivine in the Copernicus Crater generally has a weak absorption feature near 2000 nm. The suspected volcanic glass masses expose the surrounding olivine in the northwest of the crater bottom, which causes the absorption band near 1050 nm shifting toward the shorter wavelength (curve 5 in Figure 7). The olivine exposed in the south crater wall near 1050 nm and 2000 nm exhibits strong absorption features (curve 8 in Figure 7). This observation conforms to the spectral features of olivine and indicates that olivine material in the south crater wall contains few impurities. There is no uniform variation trend in the detection of olivine-dominated spectra in the central peak.

On the basis of the numerical simulation of the formation of the Copernicus Crater, Yue et al. [38] concluded that the olivine in the central peak may come from meteorite residuals. As shown in Figure 8, the error range of predicted relative Fo# is from 9 to 17 Fo# units. Calculating the average value of relative Fo# for each data point, the lowest and highest Fo# are approximately 55 and 78, with an average of 70 and a SD of 6.5. The relative Fo# of olivine in the central peak fluctuates considerably,

which indicates the coexistence of magnesian olivine and ferrous olivine. For this reason, olivine in the Copernicus Crater's central peak may not come solely from deep within the moon or even from the lunar mantle. Dhingra et al. [39] summarized three possible sources of olivine in the Copernicus impact crater: (1) Rapid cooling crystallization when deep lunar materials invade magnesian fusant, (2) existing olivine and Mg/Fe glass masses sputtering onto the crater walls on considerable impact, and (3) meteorite residuals of asteroid impact. As to second and third explanations, all exposed olivine material in the Copernicus Crater have relatively uniform Fo#. However, the Fo# of central peak olivine differs significantly, and the Fo# of the olivine in the crater walls and bottom is fairly average. Accordingly, the olivine in the crater walls and bottom was likely formed by the means proposed in the first explanation.

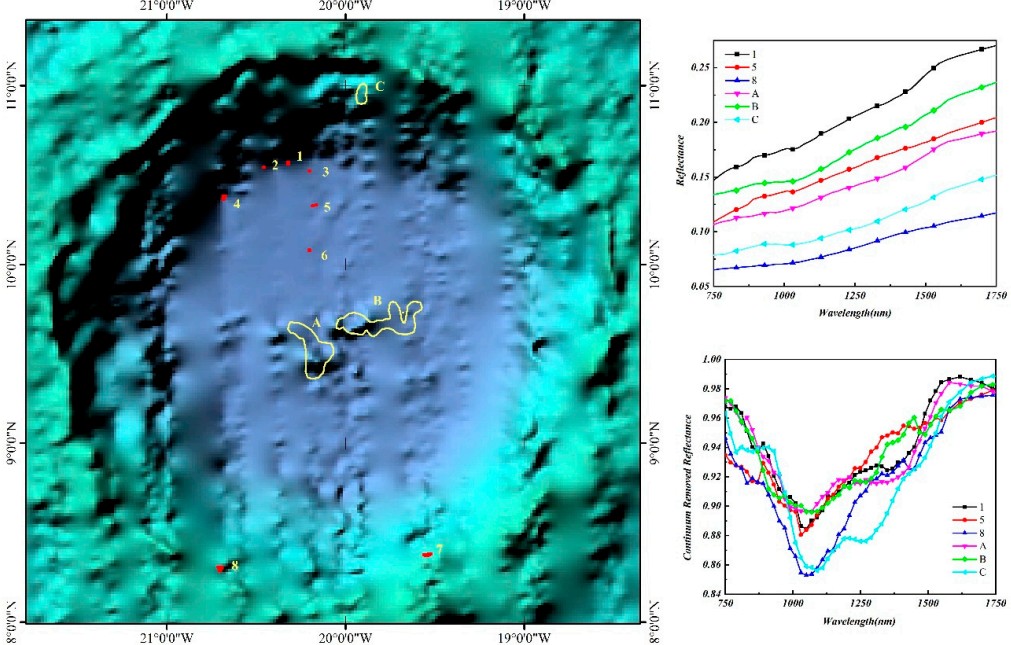

**Figure 7.** Olivine distribution in the Copernicus Crater and its spectra.

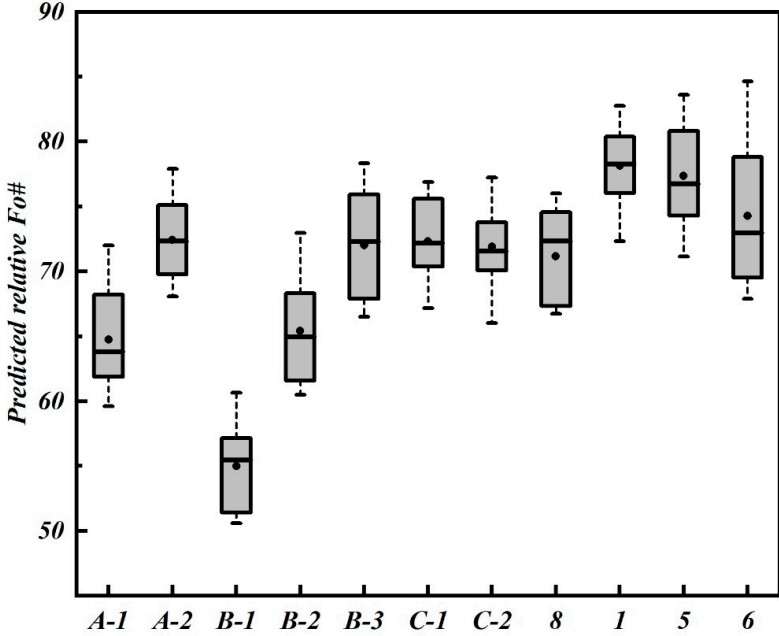

**Figure 8.** Prediction result of relative Fo# for olivine-rich materials in the Copernicus Crater.

## 5. Summary and Conclusions

This study presents a relative Fo# prediction model for lunar olivine, based on the trends between band centers and Fo# of lunar and terrestrial olivine according to the results of previous research. Olivine-dominated reflectance spectra in Sinus Iridum, the LPD dark ring in the southwest of Orientale Basin, and the Copernicus Crater are extracted using SFF based on $M^3$ data to calculate the relative Fo#. The feasibility of regarding the relative Fo# as an important additional clue of geological interpretation is discussed.

The olivine materials in the Sinus Iridum and the Copernicus Crater's walls have relatively high Fo#, which reflects early lunar deep material sources. However, the difference of their spectral characteristics can reflect their different crystallization mechanisms. The distinctive features and relative Fo# of olivine-dominated spectra in the dark ring represent the rapid early crystallization with glassy components, meanwhile constraining the geologic origin of 'OOS'. Although more regions for comparative analysis in future research should be selected, olivine Fo# calculated using the MGM did exhibit important constraint implications for disclosure of its sources.

**Author Contributions:** Conceptualization, Y.Z. and C.Z.; methodology, C.Z.; software, C.Z.; validation, C.Z., and B.Z.; formal analysis, C.Z.; investigation, C.Z. and Y.Z.; resources, S.C. and B.Z.; data curation, S.C. and B.Z.; writing—original draft preparation, C.Z. and Y.Z.; writing—review and editing, Y.Z. and S.C.; visualization, Y.Z. and B.Z.; supervision, Y.Z. and S.C.; project administration, C.Z. and Y.Z.; funding acquisition, C.Z. and Y.Z.

**Funding:** This study is jointly supported by the Key laboratory for Ecological Environment in Coastal Areas, State Oceanic Administration (201810), PhD's Research Start-up Project of National Marine Environmental Monitoring Center (2017-A-06), the Pioneer Science and Technology Special Training Program B (No. XDPB11-01-04) of Chinese Academy of Sciences, the China-Italy Collaborate Project for Lunar Surface Mapping (No. 2016YFE0104400), and National Natural Science Foundation (No. 41876109).

**Acknowledgments:** The $M^3$ data from the website of NASA Planetary Data System are highly appreciated. Authors would like to thank the editors and anonymous reviewers for their critical and helpful comments to improve the quality of this paper.

**Conflicts of Interest:** The authors declare no conflict of interest.

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
