# Peer review of "Analyzing the Magnesium (Mg) Number of Olivine on the Lunar Surface and Its Geological Significance"

_remotesensing, doi:10.3390/rs11131544_

Round 1

Reviewer 1 Report

Analyzing the Magnesium (Mg) Number of Olivine on the Lunar Surface and its Geological Significance

Chao Zhou, Yuanzhi Zhang, Shengbo Chen, Bingxue Zhu

The manuscript applies a modified empirical regression based model to estimate lunar olivine Fo# and yield information about the chemistry and maturity of the source material.  I think there is some good and interesting science in it, but I think the manuscript suffers from a couple major issues.  The biggest issue is communication and language.  In several areas the language is awkward or doesn't make sense.  Some acronyms are not defined properly in advance (or defined in the abstract and not text).   It is not always clear what the data sources are.  The captions are uninformative and lack proper source citation.  The second biggest issue with the manuscript may be a product of the first problem.  The manuscript purports to establish a lunar Fo# model, but in fact, the data and techniques are from previous publications.  The manuscript doesn't address how it is different enough to warrant that it "establishes" the model.  It does certainly re-derive and apply the model.  The manuscript doesn't discuss error, which previous publication estimated at 5-10% [ref 9 of the manuscript].

I think all of the problems can be addressed, so I pointed out a number of them below specifically.  However, in current form, in my opinion, I don't think this work is ready for publication.

Specific comments

Line 33:  Here and elsewhere, "magnesian" is more common than "magnesial"

Lines 34 and 35:  This sentence (i.e., "Influenced by magmatic rocks on the moon, Mg-rich olivine has a small density.") is a bit nonsensical to me.  Are the authors just trying to say that the Forsterite side is less dense than Fayalite side?  "Small density" is not the right term.

lines 35 and 36:  "Crystals are developed and appear first; thus, the olivine that is formed during the late period is rich in Fe but has low Fo# [3]." Again – not clear.  DO the authors mean high Fo# crystals develop in magma/melt first and take up the Mg, resulting in Fe-rich olivine in during late stage cooling (lower Fo#)?

Line 39, first sentence of second paragraph.  Hyperspectral has no defined spectral range.  They can even be U-VIS-NIR.  Maybe describe the actual instrument (e.g. line 100 to 103) and then delete the text at 100-103 and start that paragraph (at 103) with "The main....".

Line 42 to 43 – define SELENE

Line 49.  Define M3 (outside of the abstract) in the text first.

Line 59 and 60, first sentence of paragraph – this statement is not strictly true as written.  What about Mossbauer on the MER rovers?

Lines 71 and 72.  Last sentence in paragraph – "address" would be a better word choice than "analyze"

Line 76 – do the authors mean "constraining" rather than "limiting"?

Line 86 and elsewhere.  "earth olivine" is better worded as "terrestrial olivine". 

FIGURE 1.  For purposes of display in black and white, use different line patterns for each line.  Also, it is hard to see the difference between the dark blue and the black..  I think the X-axis is actually in microns?  Whenever data are coming from a different source than the actual study for a figure, they should be cited in the caption.

Paragraph, lines 90 to 97.  I think the authors need to be very clear here that the data are from Isaacson et al.  In fact, Isaacson et al. did linear fits of these data and used them.  They just did them a bit differently for the lunar samples and didn't publish the equation.  I discuss this in the general comments.

Line 97.  "...more evident."

FIGURE 2.  The data source should be cited in the caption.

Line 106.  I think the authors mean "spectral" rather than "spatial" here.

Line 135.  "1050 +/-50nm".  This range exceeds that of the model – even Isaacson's model which includes a larger range of Fo#. 

Line 135 to 137.  This sentence makes no sense as written.  In contrast to what?  Accompanying where?  It needs to be better explained.

Lines 134 through 158 (Generally):  What exactly is being called the "olivine enrichment zone" needs to be better defined.

FIGURE 4:  The X-axis should at least be labeled as "Sample Number".  I am not sure this is the best way to display these data – it was a bit misleading at first glance.  However, I think labeling the axis clarifies it and I can't think of a better way to display it right now.

Line 161 and throughout.  I think "material" is less awkward than "matters". 

Line 162.  It's the "youngest".  I would add "major" or "large" before "impact".  I think the citation is; Fassett, C. I., et al. "Lunar impact basins: Stratigraphy, sequence and ages from superposed impact crater populations measured from Lunar Orbiter Laser Altimeter (LOLA) data." Journal of Geophysical Research: Planets 117.E12 (2012).

Line 163.  It's the Rook mountains I believe (or Montes Rook), not Luke.  Actually, the feature in Figure 5 is at Rimae Focas.

Lines 164 to 167, - sentence about Head et al.'s work.  It is a bit confusing as written.  I think what Head was trying to say (if I recall correctly) is that the collapse feature in the center of the ring is the source of the dark material.  The placement of the dark material excavated the subsurface, and resulted in a collapse feature at the center of the ring.

Line 168.  "most possible" I think should be "most plausible".

Lines 170 to 172.  The last 2 sentences of this paragraph don't make sense as written.  For example, in "The minerals in the dark ring are domainated (spelling) by pyroxene and are distributed in the region 98° W east of the dark ring, particularly in the northwest corner of the dark ring and the east of the central peak." there are several issues.  There should be a comma after "98° W".  However, even with that, 98° W is within and on the west side of the "dark ring".  Do the authors mean "East of 98° W, the minerals in the dark ring are dominated by pyroxene..."?  Also, there is no central peak (see SELENE image provided).  This is an extrusive feature, not an impact feature.  The authors indicate they know this later in the manuscript, so maybe some other confusion has occurred.

In, "Low olivine content mainly occurs in the region 99° W west of the dark ring (Figure 5)."  The sentence might read better as "Low olivine content mainly occurs west of 99° W of the dark ring (Figure 5)." However, both sentences seem to be indicating poor olivine content – one region is mainly pyroxene and the other is low in olivine content.  Is that what the authors want to say?

FIGURE 5.  Need more details in the captions and definitions of symbols.

Line 181.  "glassy" is a better word choice.

Line 195.  I think "late" is better than "youngest" since Copernican is the current-day period. Also, the geologic age of the carter *is* ~0.8 Ga (not was).

FIGURE 8.  Needs and X-axis.

Line 226.  I don't think this study "establishes" the model as much as it calculates and applies it.  Isaacson [ref 9] did the same thing with the data same data.  They seemed to stop short of publishing the fit (I would review that paper to confirm this), potentially because they could not get a good handle on deviations from Sunshine, J.M., Pieters, C.M., (1998. Determining the composition of olivine from reflectance spectroscopy. J. Geophys. Res. 103, 13675–13688) terrestrial work because the lunar data all come from only two locations (in their interpretation).  I think saying "establishes" the model is a big overstatement.  But the manuscript does apply it, and I do not believe Isaacson et al. did that.

Lines 231 to 233.  To tightly tie the results of the study to stratigraphic age based on so little location data I think is an overstatement.  In line 232, change "can reflect" to "may reflect".  I think that softens the statement enough.

Author Response

Dear Reviewer,

Thank you for your helpful comments to improve our manuscript's quality.

We have made a careful revision according to your suggestions and comments.

Resubmitted is the revised version of our manuscript (496594) and our responses to your comments.

Your sincerely,

Authors 

Reviewer 2 Report

This article concerning the use of remote sensing of hyperspectral imaging of olivine from the lunar surface to interpret geological evolution on the moon. Generally this is an interesting topic and would be suitable for publication. However, I see many potential flaws and areas for concern in this manuscript which give me cause for hesitation. I have two major concerns which are detailed below. Ultimately I recommend rejection of the article. I think the authors need to go back to the drawing board and rethink the research from the ground up. If these concerns can be addressed then I think this manuscript could be resubmitted and it might ultimately be a very good contribution.

-Figure 2: The data contained in Figure 2 is the basis for the methodology developed in this manuscript. However, there is almost no information on how these data were collected. For this to work the Mg number (Fo#) would have had to have been determined independently from the reflectance data. Ideally this would be done using Electron Probe Microanalysis (EPMA) as this is the most accurate method. But the authors completely gloss over this point. Also, how were the reflectance data collected? Presumably these are physical samples the authors have in the lab and so the reflectance data was collected by another method than the remote sensing data collected in the rest of the paper... This needs to be clearly spelled out in a methods section.

-Sources of error and impact on geological interpretations:

The data in Figure 2, upon which the manuscript is based, show a strong dependence of Fo# on the wavelength of the absorption lines. Even very small variations in the wavelength can result in large differences in inferred Fo#. Therefore, even small errors associated with measuring the band positions will result in very large errors on the inferred Fo#. This can be seen even for the calibration data in Figure 2 in which many of the points fall off the linear regression and would have very different measured Fo# inferred from the reflectance spectra than their true measured Fo# values. One glaring omission from this manuscript is any statement of measurement errors on the measured absorption wavelengths. Looking at the remote sensing spectra from Figure 3, 5, and 7, frankly some of these spectra are very low quality and I would have to assume there are significant errors associated with measuring the positions of the m1-1, m2, and m1-2 bands. The authors need to quantify these errors somehow and reject any data with errors that are too large to allow for reasonable geological interpretation. The authors claim there are differences in Fo# and interpret these differences geologically, but how much of this variation is due to error in the measurement of the absorption bands? I'd have to guess that many of the spectra have significant error. This issue is critical and needs to be addressed before this manuscript can be considered for publication.

Author Response

(The authors gave the same response as above.)

Reviewer 3 Report

A major concern of this work is that continuum removal plays a big role on deriving the Mg#. However, the method of continuum removal was not included in the manuscript. And the continuum removed spectra in the manuscript do not look right. This needs to be fixed.

Caution should be taken for performing continuum removal, because the current M3 L2 reflectance data show significant thermal residuals at longer wavelength (i.e., > 2.5 microns) (Li & Milliken, 2016; Bandfield et al., 2018). One can either re-do the thermal correction of M3 data with improved methods or cut off current M3 data at 2.5 microns to work around the thermal issue.

Another comment is that the author did not mention any of previous work about mapping lunar surface Mg# from orbital data, such as Lucey, 2004; Ohtake et al., 2012; and Crities and Lucey 2015. These work should be mentioned in the manuscript and the difference and improvement of this work should be clarified.

Since the same method MGM is used as that of Sunshine and Pieters 1998 and Isaascon et al., 2010, the difference and improvement of this work should also be clarified.

Other minor comments are annotated in the PDF file.

Author Response

(The authors gave the same response as above.)

Round 2

Reviewer 1 Report

This is my second time seeing this manuscript, Analyzing the Magnesium (Mg) Number of Olivine on the Lunar Surface and Its Geological Significance by Chao Zhou et al.  I find this version much improved.  It is far easier to read and understand.  There are still some outlying issues however, and I note them below.  Nearly all of these are typographical and those that are not, are minor.  Outside of typographical errors, my most import comment is my last (re line 346).  This may sound like a style comment, but in fact it is more than that.

General comments

There may be some structure problems.  For example, while there are methods - there isn't a called out "methods" section.  I think simple heading changes can address this.  I defer here to the editor.

Specific comments - these line numbers apply to the marked up version

Line 38 remove "diapiric".  Or, respell as "diapirically"

Line 42 comma after "Accordingly"

Line 60 The Editor will correct me if I am wrong, but Mneeds to be defined in the text first - i.e. "...Moon Mineralogy Mapper (M3)..." - just like in line 65 for LSCC.  I understand it was already done in the abstract, however, in general the first time acronyms or abbreviations appear in the body of the manuscript this should also be done.  I made a comment in my initial review on the subject, but I think it was not clear.

Line 68 Typo - "This observation is not consistent..."

Line 69, only 1 "mainly" is required (delete one of them)

Line 74 typo - should be "provides"

Line 75 "...from ray spectrometer data,..." ,  gamma-ray?  If so, it should say gamma-ray.  If not, I would just remove the word "ray"

Line 81 typo - "influenced" should be "influence"

Line 82 "of" should be removed

Line 83 remove "was"

Line 83 to 85 - sentence "... calculates the main parameters of the characteristic absorption peak according to of substances spectrum, including...", should be, "... calculates the main parameters of the characteristic absorption peak according to features of the spectrum, including..."

Line 133  "ways" is the wrong word.  use something like feature, attribute, trait, trend, or quality.

Line 135 "distances" should be "distance"

line 135 "uses" should be "used"

Line 159 there should be a comma after "method" - there may already be one and I can't see it because of the markup.

Line 184  Something does not read right.  I think "20" can be removed.

lines 188 to 190 - "In order to avoid the errors are too large to allow for reasonable geological interpretation, the larger error are eliminated that greater than 20 Fo# units." does not read properly.  Do the authors mean, " In order to avoid errors that are too large for reasonable geological interpretation, larger errors of greater than 20 Fo# units are eliminated."

Line 253, remove "exists" (redundant)

Line 257, period needed at end of sentence.

Line 263, "As shown" can be removed. 

Line 264, "area" should be "areas".

Line 269, "OOS" does need to be defined, e.g. "...olivine–orthopyroxene–spinel (OOS)..."

Lines 270 and 271, " which were composed three distinct rock type dominated by orthopyroxene, olivine and spinel"  can be removed (redundant)

Figure 5.  Caption should note the overlapping lines in reflectance plot.  C is over B, Oli is over A, etc...  I'm unsure of any way to better clarify this.

Regarding "the Copernicus crater" vs "Copernicus Crater".  I mention this only because it appears the authors were trying to address it.  In my form of English (American), both are correct, however, "the Copernicus crater " sounds awkward as one is normally talking about a location, not the object.  Capitalization of crater after a name is optional (but must be consistent - see line 337).  However, there may be an editorial preference I am not aware of.

Line 346 - I still do not like the word "establishes" here for reasons in my original review - I think "presents" is better and safer.  I do understand the authors have been very clear about the modeling source/history, but "presents" avoids any potential confusion.  The word "establish" implies "the very first" and I think that invites argument in this case.

Author Response

Dear Reviewer,

Thank you for your comments. We have revised and improved the revision based on your comments.

Yours sincerely,

Yuanzhi Zhang

Reviewer 2 Report

The authors have done a good job of considering and incorporating comments from all the reviewers. I have no further comments or suggestions, I think the manuscript is acceptable to publish in its current form.

Author Response

Dear Reviewer,

Thank you for your comments. We have make the English checking.

Yours sincerely,

Yuanzhi Zhang

Round 3

Reviewer 1 Report

I believe the authors have addressed any comments and concerns I have.  This version is much improved over the initial. 

Author Response

Dear Reviewer,

Thanks for your comments to improve the quality of the manuscript.

Yours sincerely,

Authors